# Ongoing Challenges with Healthcare-Associated *Candida auris* Outbreaks in Oman

**DOI:** 10.3390/jof5040101

**Published:** 2019-10-23

**Authors:** Amal Al Maani, Hema Paul, Azza Al-Rashdi, Adil Al Wahaibi, Amina Al-Jardani, Asma M. Ali Al Abri, Mariam A. H. AlBalushi, Seif Al Abri, Mohammed Al Reesi, Ali Al Maqbali, Nashwa M. Al Kasaby, Theun de Groot, Jacques F. Meis, Abdullah M. S. Al-Hatmi

**Affiliations:** 1Directorate General for Disease Surveillance and Control (DGDSC), Ministry of Health, 393 Muscat, Oman; adilwahaibi@gmail.com (A.A.W.); salabri@gmail.com (S.A.A.); 2Department of Infection Prevention and Control, DGDSC, Ministry of Health, 393 Muscat, Oman; hemapaul23@gmail.com; 3Central Public Health Laboratories, DGDSC, Ministry of Health, 393 Muscat, Oman; a.rashdi24@gmail.com (A.A.-R.); aksaljardani@gmail.com (A.A.-J.); 4Infection Prevention and Control Department, Sohar Hospital, Ministry of Health, 749 Sohar, Oman; alabri.asma@gmail.com (A.M.A.A.A.); Ipcsuhar@gmail.com (M.A.H.A.); 5Paediatric Infectious Diseases Unit, Sohar Hospital, Ministry of Health, 749 Sohar, Oman; alreesimohammed@gmail.com; 6Department of Diseases Surveillance and Control, Directorate General of Health Services, North Batinah Governorate, Ministry of Health, 749 Sohar, Oman; almoqbali96@hotmail.com; 7Microbiology Section, Pathology Department, Sohar Hospital, 749 Sohar, Oman; nashwakasby2003@yahoo.com; 8Department of Medical Microbiology and Immunology, Faculty of Medicine, Mansoura University, 56 Mansoura, Egypt; 9Department of Medical Microbiology and Infectious Diseases, Canisius Wilhelmina Hospital, 6532 SZ Nijmegen, The Netherlands; t.groot@cwz.nl (T.d.G.); jacques.meis@gmail.com (J.F.M.); 10Centre of Expertise in Mycology Radboud University Medical Centre/Canisius Wilhelmina Hospital, 6532 SZ Nijmegen, The Netherlands; 11Westerdijk Fungal Biodiversity Institute, 85167 Utrecht, The Netherlands; 12Ministry of Health, Directorate General of Health Services, Al-Dhahirah Governorate, 46 Ibri, Oman

**Keywords:** *Candida auris*, infection, outbreak, Sohar Hospital, Oman

## Abstract

*Candida auris* has emerged in the past decade as a multi-drug resistant public health threat causing health care outbreaks. Here we report epidemiological, clinical, and microbiological investigations of a *C. auris* outbreak in a regional Omani hospital between April 2018 and April 2019. The outbreak started in the intensive care areas (intensive care unit (ICU), coronary care unit (CCU), and high dependency unit) but cases were subsequently diagnosed in other medical and surgical units. In addition to the patients’ clinical and screening samples, environmental swabs from high touch areas and from the hands of 35 staff were collected. All the positive samples from patients and environmental screening were confirmed using MALDI-TOF, and additional ITS-rDNA sequencing was done for ten clinical and two environmental isolates. There were 32 patients positive for *C. auris* of which 14 (43.8%) had urinary tract infection, 11 (34.4%) had candidemia, and 7 (21.8%) had asymptomatic skin colonization. The median age was 64 years (14–88) with 17 (53.1%) male and 15 (46.9%) female patients. Prior to diagnosis, 21 (65.6%) had been admitted to the intensive care unit, and 11 (34.4%) had been nursed in medical or surgical wards. The crude mortality rate in our patient’s cohort was 53.1. Two swabs collected from a ventilator in two different beds in the ICU were positive for *C. auris*. None of the health care worker samples were positive. Molecular typing showed that clinical and environmental isolates were genetically similar and all belonged to the South Asian *C. auris* clade I. Most isolates had non-susceptible fluconazole (100%) and amphotericin B (33%) minimal inhibitory concentrations (MICs), but had low echinocandin and voriconazole MICs. Despite multimodal infection prevention and control measures, new cases continued to appear, challenging all the containment efforts.

## 1. Introduction

*Candida auris* is an emerging multidrug resistant human pathogenic yeast causing bloodstream infections mainly in hospitalized patients [1,2,3]. It was identified a decade ago and since then was isolated from patients in at least 35 countries worldwide including the Middle East and Oman [4,5,6,7,8,9,10,11]. Health care outbreaks due to *C. auris* have been reported with thirty-day crude mortality ranging from 28% to 50% [12,13,14,15]. The first outbreak of *C. auris* infection involving 12 patients was reported from India between 2009 and 2012 [16], followed by the first in Europe involving 50 patients in London, between April 2015 and July 2016 [17]. The largest outbreak took place in Spain between April 2016 to January 2017 where there were 41 confirmed blood stream infections, of which five patients developed septic complications, and 140 others were colonized [18]. In the United States, the first seven cases of *C. auris* occurred between May 2013 and August 2016, reported retrospectively [19]. At the end of August 2019, the CDC reported more than 2000 clinical and colonization cases of *C. auris* from all over the US. (https://www.cdc.gov/fungal/candida-auris/tracking-c-auris.html) [20].

The control of *C. auris* healthcare-associated outbreaks has been a challenge to the affected facilities, which resulted in a building up of cases from different regions of the world including those with well-established diagnostic, management, and preventive strategies [1,15,17,21]. The main reasons hindering control of such outbreaks are the difficulties in identification, unknown population prevalence, the uncertain environmental niches, and the unclear mechanisms of spread [17,19,22]. A recent review has suggested that the implementation of broad-ranging infection prevention and control care bundles was shown in limited studies to be effective at reducing the number of invasive disease but its impact on rate of colonization was unclear [1]. We describe the first nosocomial outbreak of *C. auris* with genetically related environmental and clinical isolates in Sohar city, Oman, occurring between April 2018 and April 2019. 

## 2. Materials and Methods

### 2.1. Setting

The outbreak occurred in a secondary care regional (363-bed) hospital in Oman (Sohar Hospital) with different medical and surgical units. The intensive care department is composed of two units; the intensive care unit (ICU), which is an open multi-bed space that can accommodate up to 6 ventilated patients (medical and surgical cases), and the coronary care unit (CCU), which has capacity for 8 patients in single rooms. The department is staffed by 3 intensivists, 30 nurses, and 4 respiratory technicians. All the surgical and medical wards are multi-bed (6 cubicles) with shared toilet facilities and some have one or two single isolation rooms. The capacity of each cubicle is 4–6 beds which are separated by curtains keeping a minimal space of 1.2 m between them. Each patient will be attended to by a caregiver who will be usually occupying the space between the patients’ beds. 

### 2.2. Case Definition

Any patient admitted to the facility for more than 48 h with a positive screening or clinical sample culture for *C. auris* was included. The case was stratified as infection versus colonization based on clinical and/or laboratory markers as an evidence of infection.

### 2.3. Isolation of C. auris

*Candida auris* was isolated by routine microbiology procedures from clinical samples, such as blood, urine, surgical wounds, and catheters, as well as from screening samples (axilla, groin, nasal, throat, and perianal swabs).

### 2.4. Sample Collection from Environment and Health Care Workers

Environmental samples were collected from the ICU and high dependency units especially from high touch areas and re-useable devices (including ventilators, patient’s cots, steel trolleys, staff and patient lockers, glove boxes, bed lights, racks, sinks, tissue boxes, hand rub dispensers, curtains, floors, head lights, and infusion pumps), using sterile swabs moistened with sterile saline. The first set of environmental samples were collected on 9 October 2018, just before conducting cleaning using a chlorine-based disinfectant (1% sodium hypochlorite with 16.5% sodium chloride) and environmental decontamination with hydrogen peroxide (H_2_O_2_) fumigation. Another set of swabs were collected after cleaning and decontamination of the ICU on 16 October 2018. A total of 140 environmental swabs were collected for screening purposes. Hand swabs were also collected from 35 health care workers (HCWs) in intensive care and high dependency areas. This included nurses, intensivists, respiratory therapists, and medical assistants. All swabs were transported to the laboratory in charcoal transport medium and were inoculated on Sabouraud dextrose agar (SDA) and incubated at 37 °C for 48 h.

### 2.5. Laboratory Investigations

All the positive *Candida* isolates processed in the hospital laboratory (clinical or screening) were referred to the Central Public Health laboratory (CPHL) for confirmation. The isolates were preliminary identified by phenotypic and biochemical characteristics. Biochemical characteristics were analyzed by API 20C AUX (bioMerieux). Further identification was done using MALDI-TOF MS (Maldi Biotyper MBT Smart V8.0.0.0.7171, Bruker Daltoniks, Bremen, Germany) at Canisius Wilhelmina Hospital, Nijmegen, The Netherlands [23]. There were 12 *C. auris* isolates (2 environmental and 10 clinical) examined for further identification and molecular typing at Canisius Wilhelmina Hospital, Nijmegen, The Netherlands.

### 2.6. Identification Using PCR-Sequencing and Genotyping

Strains were transferred to fresh glucose–yeast–peptone agar (GYPA) plates and incubated at 25 °C for 48 h. DNA extraction was performed by the Quick CTAB (cetyltrimethylammonium bromide) extraction method according to the protocol described by Al-Hatmi et al. [24]. Briefly, approximately 1 full loop of 48 h cultures was transferred to a 2 mL screw-capped tubes filled with a 490 μL 2% cetyltrimethylammonium bromide (CTAB) buffer and 6–10 acid-washed glass beads (diameter 1.5–2.0 mm, Sigma). A total of 10 μL Proteinase K were added and mixed on a MoBio vortex for 10 min, and the mixture was incubated at 60 °C for 30 min. After incubation, 500 μL chloroform:isoamylalcohol (24:1) was added and shaken for 2 min. The tubes were centrifuged for 10 min at 14,000 rpm, supernatants were collected in new 1.5 mL Eppendorf tubes, and ~270 μL of ice-cold iso-propanol was added followed by centrifugation again at 14,000 rpm for 10 min. Pellets were washed with 1 mL ice-cold 70% ethanol, dried using a vacuum dryer, and re-suspended in 50 μL TE-buffer. DNA concentrations were measured with a NanoDrop 2000 spectrophotometer (Thermo Fisher, Wilmington, NC, USA). Extracted DNAs were stored at −20 °C until use.

PCR amplification and sequencing of the rDNA internal transcribed spacer region (ITS) was performed using primers ITS4 and ITS5. The internal transcribed spacer region (ITS) was amplified. The PCR reactions mixture contained 1 μL template DNA (50 ng), 1.25 μL 10× PCR buffer, 1 μL dNTP mix (2.5 mM), 0.5 μL of each primer (10 pmol), 0.2 μL Taq polymerase (Biotaq, Bioline, Germany) (5 U/μL), BSA 0.5 μL, and water to complete the final volume of 12.5 μL. PCR was performed in an ABI PRISM 2720 (Applied Biosystems, Foster City, CA, USA). Amplifications were performed as follows: 95 °C for 5 min, followed by 35 cycles consisting of 94 °C for 45 s, 52 °C for 45 s and 72 °C for 2 min, as well as a post-elongation step at 72 °C for 7 min. PCR products were visualized by electrophoresis on a 1% (w/v) agarose gel. Sequencing reaction mixtures contained 1 µL template DNA (0.1 pmol), 1 µL primer (4 mM), 1 µL of BigDyeTM terminator (Applied Biosystems), 3 µL buffer, and 5.5 µL ultra-pure water to 10 µL final volume. Sequencing PCR was performed as follows: 95 °C for 1 min, followed by 30 cycles consisting of 95 °C for 10 s, 50 °C for 5 s, and 60 °C for 2 min. DNA sequences were edited and consensus sequences were assembled by the SeqMan package of Lasergene software (DNAStar, Madison, WI, USA). Sequences were exported as FASTA files. For preliminary identification, a homology search for the sequences of ITS was done using the BLAST tool in NCBI and CBS databases down to species level. For conclusive identification, sequences were aligned with MAFFT (www.ebi.ac.uk/Tools/msa/mafft/), followed by manual adjustments with MEGA v. 6.2 and BIOEDIT v. 7.0.5.2. The ML trees were constructed with MEGA v. 6.2. Maximum likelihood (ML) analysis was done with RAxML-VI-HPC v. 7.0.3 with non-parametric bootstrapping using 1000 replicates. For genotyping *C. auris* a small tandem repeat (STR) analysis was employed [25]. Briefly, DNA was extracted and purified with the MagNA Pure LC instrument and the MagNA Pure DNA isolation kit III (Roche Diagnostics GmbH, Mannheim, Germany), according to the recommendations of the manufacturer. Strains were resuspended in 50 μL physiological salt and after addition of 200 U of lyticase (Sigma-Aldrich, St. Louis, MO, USA) and incubation for 5 min at 37 °C, 450 μL physiological salt was added. The sample was then incubated for 15 min at 100 °C and cooled down to room temperature. Four multiplex PCR reactions, which amplify 12 STR targets with a repeat size of 2, 3, or 9 nucleotides, was done for genotyping the Omani isolates, which were compared with the global collection [25]. Copy numbers of the twelve markers of all isolates were determined using GeneMapper Software 5 (Applied Biosystems). Relatedness between isolates was analyzed using BioNumerics v. 7.6.1 software (Applied Maths, Kortrijk, Belgium) via the unweighted pair group method with arithmetic averages, using the multistate categorical similarity coefficient.

### 2.7. Antifungal Susceptibility Testing

In vitro antifungal susceptibility testing of a selection of *C. auris* isolates (*n* = 12) was performed using the M38-A2 broth microdilution method of CLSI (Clinical and Laboratory Standards Institute) [26].

### 2.8. Statistical Analysis

Descriptive statistics of the cases were presented as the mean, the median for continuous variables, and percentages for categorical variables. These analyses along with the chart for patient transfer within the hospital were constructed using Excel software.

## 3. Results

### 3.1. Patients’ Cohort Description

The first case of candidemia during this outbreak was identified in June (week 21) 2018 after a patient died (Figure 1). A total of 32 patients were identified (25 infected and 7 colonized) with *C. auris* from April 2018 to April 2019 (Table 1). There were 11 (34.4%) patients with candidemia and 14 (43.8%) with urinary tract infection. The affected age group ranged from 14–88 years with a median of 64 years. The male to female ratio was 17:15. A total of 75% (*n* = 24) of affected patients had comorbidities that included diabetes mellitus, hypertension, and cardiovascular, neurological and immunodeficiency diseases. Antibacterial usage prior to diagnosis was reported for 16 (50%) patients, and 20 (62.5%) patients were on antifungals including fluconazole, voriconazole, and lipid-based formulations of amphotericin B.

Among the infected cohort, 20 (63%) patients had been admitted to the ICU prior to the diagnosis but 11 (34.4%) were in other medical or surgical wards (Figure 1). The mean number of admission days prior to developing infection was 28.4. Out of the 32 infected patients 17 died (53.1%), 12 (37.5%) recovered, and 3 (9.3%) were still admitted (Table 1).

### 3.2. Outbreak Timeline

The first case of candidemia during this outbreak was identified in June (week 21) 2018 after the dead of a patient. Cases continued to accumulate intermittently (2–6 weeks of no new cases) but transmission was ongoing up to April 2019. The epidemiological curve (Figure 2) shows the incidence of new cases along with accumulated number of admitted positive patients during each week of the year. The main event from April 2018 when the first case was admitted until the end of April 2019 is depicted. The same hospital had a Middle East Respiratory Syndrome Coronavirus outbreak (MERS-CoV) outbreak at the end of January 2019 (week 5), which had resulted in activating the hospital’s emergency plan. Elective admissions to the hospital were postponed, restricting visitors to isolated patients, evacuating patients from intensive care units and medical wards to an isolation unit while conducting terminal cleaning and fumigation with H_2_O_2_, transferring stable emergency cases to other hospitals, and conducting daily infection control rounds to observe health care worker adherence to hand hygiene and care bundles. The above measures resulted in ending the MERS-CoV transmission but new cases of *C. auris* continued to occur. 

### 3.3. Mycology

Among the different isolates collected during this outbreak, 32 were found to be positive after removing duplicates from the same patient. There were seven patients who had *C. auris*, recovered from more than one site at different episodes during their stay in the hospital. Out of 140 swabs collected from environmental surfaces of ICU, CCU, and high dependency areas, two swabs from the ICU were found to be positive for *C. auris* (ventilator in bed No. 6 and a steel trolley near bed No. 5 in the ICU). Repeated swabs after environmental cleaning in the ICU were negative. None of the samples collected for screening from health care workers grew *Candida*. The *C. auris* strains were identified by MALDI-TOF MS for *Candida* species with confidence log scores > 1.92, indicating correct generic and species identification.

### 3.4. Molecular Identification and Genotyping

A part of the ITS region was used for identification of positive clinical and environmental isolates. ITS gene possessed enough polymorphisms and, therefore, was an excellent marker with 99%–100% accuracy for the identification of *Candida* species to be *C. auris*. For more accurate identification, the phylogenetic position of *C. auris* was established using Maximum likelihood (ML) analysis with RAxML-VI-HPC v. 7.0.3 (500 bp) for the ITS region. *Candida* species within the *C. haemulonii* complex and other closely related *Candida* species were selected for phylogenetic analyses and sequences of the ITS gene were aligned among the sequences available from GenBank. The ITS phylogenetic analyses showed that the reported clinical isolates were *C. auris* and were found to be identical to many other clinical strains of *C. auris* from all over the world (Figure 3). By using STR genotyping, the isolates from Oman clustered with Indian isolates in the South-Asia clade 1 (Figure 4) while isolates from South Africa, Japan/Korea, Venezuela, and Iran each clustered in the other four major *C. auris* clades, previously identified via whole-genome sequencing (WGS) [4,27]. 

### 3.5. Antifungal Susceptibility

Table 2 summarizes the MIC values of eight antifungal drugs against *C. auris* (*n* = 12). There was a uniform pattern of MICs below the ECOFF for itraconazole, voriconazole, posaconazole, isavuconazole, anidulafungin, and micafungin [27]. The highest non-susceptible MICs were recorded for fluconazole (MIC_50,90_ > 64 mg/L) and amphotericin B (MIC_50_ 1 mg/L, MIC_90_ 2 mg/L).

### 3.6. Infection Prevention and Control Measures

With the increasing incidence of candidemia in the intensive care unit, with high mortality, an outbreak team was formed to investigate and manage the situation (Figure 2). A retrospective search of one year prior to diagnosis of the first case in the hospital microbiology data base showed no previous isolates of *C. auris*. Some actions were immediately taken, including: creating awareness about *C. auris* among the health care workers; training all staff of involved units on standard and contact isolation precautions; involving an administration and infection control committee; regular infection control rounds for the affected units, with monitoring of adherence to hand hygiene and other precautions with immediate feedback to staff; implementing screening for *C. auris* on routine admission to the ICU and all contacts of positive cases; and ensuring proper terminal cleaning of rooms and bed spaces. There had been a period of six weeks where no further new cases occurred (weeks 27–32) but starting from week 33 onwards new cases were diagnosed from different units and wards (Figure 1 and Figure 2). Environmental and health care worker screening was done in the ICU where most patients had been identified. This was followed by extensive environmental cleaning with infection control supervision. Cleaning was done with a chlorine-based disinfectant (1% sodium hypochlorite with 16.5% sodium chloride), in a ratio of seven tablets per 1.5 L of water. This was used for beds and all articles surrounding them and for mopping the floor. Different mops were used for each cubicle and aldehyde free alcohol wipes were employed to clean all equipment and monitors. All positive cases were isolated with designated medical equipment and toilet facilities. Extensive environmental cleaning and decontamination was done with fumigation with H_2_O_2_. In addition, the implementation of care bundles in all units, strictly enforced adherence to infection control measures, and instituting an antimicrobial stewardship program in the hospital, as well as a competency-based certification course in infection control for all HCWs were conducted. All the above measures resulted in earlier detection of cases and improving the outcome of patients, but the incidence of new cases had continued (Figure 1).

## 4. Discussion

We report to the best of our knowledge the largest outbreak in the Middle East region from one facility with 32 patients affected and a crude mortality rate of 53%. In the last three years, six countries in the Middle East, including Oman (Iran, Kuwait, KSA, UAE, and Israel) have reported *C. auris* in adult patients with several underlying comorbidities [28,29]. 

The 30-day crude mortality rates with *C. auris* infection have been variable in different geographical regions and found to range between 0% and 72% [1]. A recent review from the Middle East showed a crude mortality rate of 60% but in the United Kingdom no deaths were directly attributable to *C. auris* infection. Our series showed a crude mortality rate of 53.1%. The high crude mortality maybe a reflection of the comorbidities within the infected population and the diagnostic delay, in addition to the resistant nature of the pathogen [17]. 

*Candida auris* is known to affect ICU patients, especially those with medical devices (CVCs, urinary catheters, etc.). The latter suggests a potential role for biofilm formation [13,17]. The role of environmental contamination in facilitating *C. auris* transmission, illustrated in this outbreak, confirms previous similar findings elsewhere, including the Oxford University Hospitals outbreak where sequenced isolates from reusable equipment were genetically related to isolates from patients [17,30]. Environmental screening can be problematic and not cost effective because of transient, sporadic contamination. For example, a report from an ICU in India reported less than 10% environmental contamination [22]. The same is true for health care workers screening, which usually result in a large number of samples and a very low positivity rate (less than 5% in the Indian [31] and UK studies [15] and zero positive out of 35 screened HCWs in our study). The single positive HCW in the UK study cared for only one patient, who was heavily colonized with *C. auris* but was not implicated in any onward transmissions, which further questions the practical implication of screening health care workers [17]. The intensive care units had been hotspots for *C. auris* with a great challenge in breaking the vicious cycle of environment/workers contamination–patient contamination infection [32]. 

*Candida auris* can persist on different types of surfaces, including moist, dry, and plastic surfaces, with the potential of survival for up to 14 days [33,34,35,36,37]. The success of environmental decontamination was variable in the literature utilizing mostly materials and methods tested earlier against resistant Gram-positive and Gram-negative pathogens [17,21,38,39]. The use of high-concentration chlorine solutions in combination with hydrogen peroxide vapor or UV light for terminal decontamination has been suggested to be effective in controlling the transmission in some reports [17,19,21,40]. The same protocol without UV light was used in our setting twice including when it was used for controlling a MERS-CoV outbreak, but this failed to have impact on the incidence of new *C. auris* cases. This could be due to a hidden colonized population or environmental reservoir. 

The genome analysis of *C. auris* suggests that there are between 6500 and 8500 protein-coding sequences, including those for virulence factors such as biofilm formation and acquisition of drug resistance [41,42]. There is also widespread variation between geographic clades, with thousands of single nucleotide polymorphism (SNP) differences. At present, *C. auris* is separated into five geographic clades: The South Asian, African, South American, Iranian, and East Asian clades [8,39,41,43]. Phylogenomic analysis using ITS sequences and STR genotyping show that the Omani isolates belong to the South Asian clade. Most of the Omani isolates of *C. auris* exhibited non-susceptible fluconazole MICs (≥64 mg/L). Some strains of *C. auris* can be non-susceptible to multiple antifungal classes, severely limiting treatment options and making infection control and prevention guided by rapid detection in healthcare settings essential [42,43,44,45]. 

The known characteristics of *C. auris* (rapid acquisition and spread within affected facilities, high mortality rates, challenging environmental decontamination, and high levels of antifungal resistance) underscore the importance of rapid containment of the spread of this public health pathogen, including developing rapid and accurate diagnostic tools at the point of care [20,44] and investing in infection control and antimicrobial stewardship programs. A high mortality rate and the on-going challenges for containment have been highlighted, with a call for global attention to this growing antimicrobial resistant pathogen, and active research acceleration on intervention modalities.

## Figures and Tables

**Figure 1 jof-05-00101-f001:**
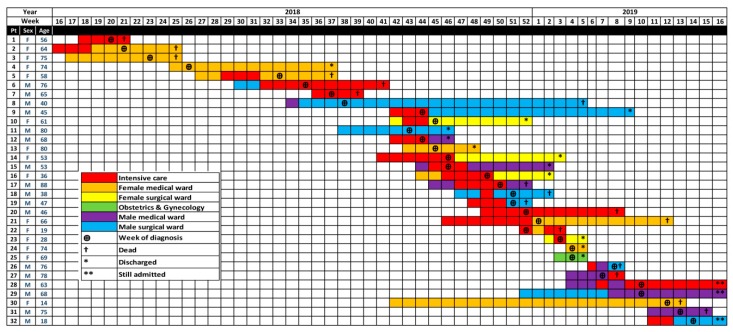
Patient location and transfer within the hospital in relation to diagnosis and outcome time (weeks of year) during the outbreak.

**Figure 2 jof-05-00101-f002:**
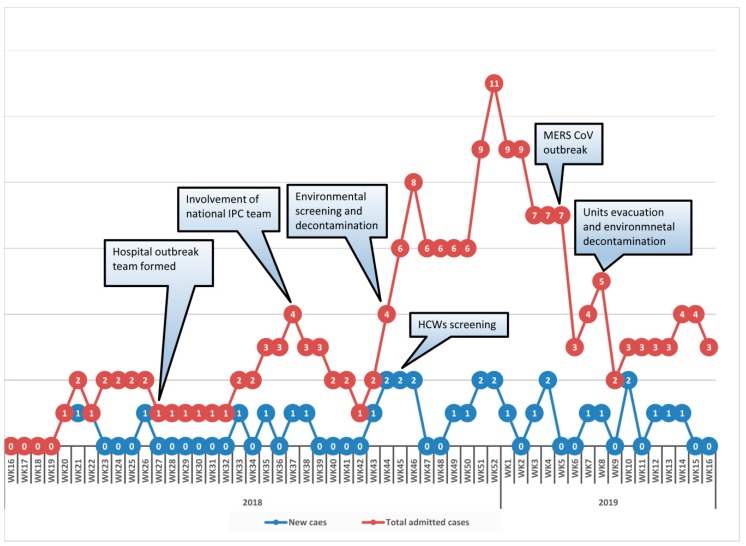
Epicurve of *C. auris* cases in Sohar hospital (April 2018–April 2019).

**Figure 3 jof-05-00101-f003:**
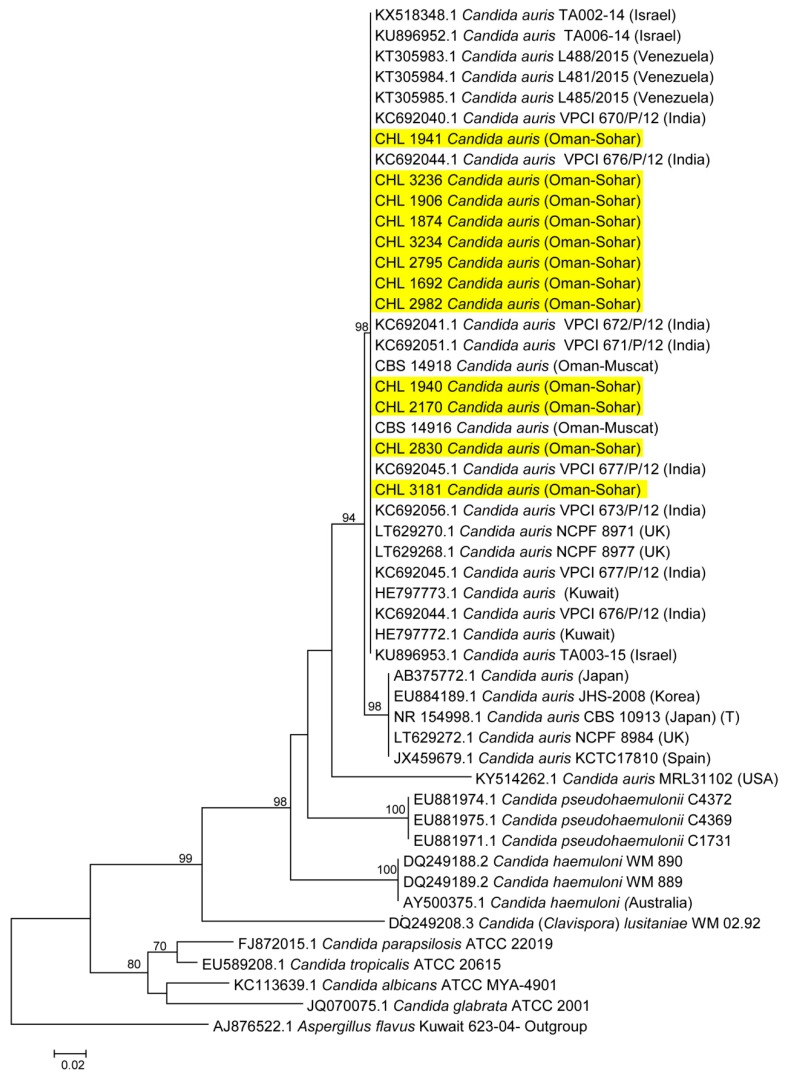
Phylogenetic tree generated by MLH analysis using ITS sequences of the *C. auris* strains with closely related *Candida* species. Bootstrap-supported values above 70% are indicated at the nodes. Yellow color indicates *C. auris* strains examined in this study from Oman.

**Figure 4 jof-05-00101-f004:**
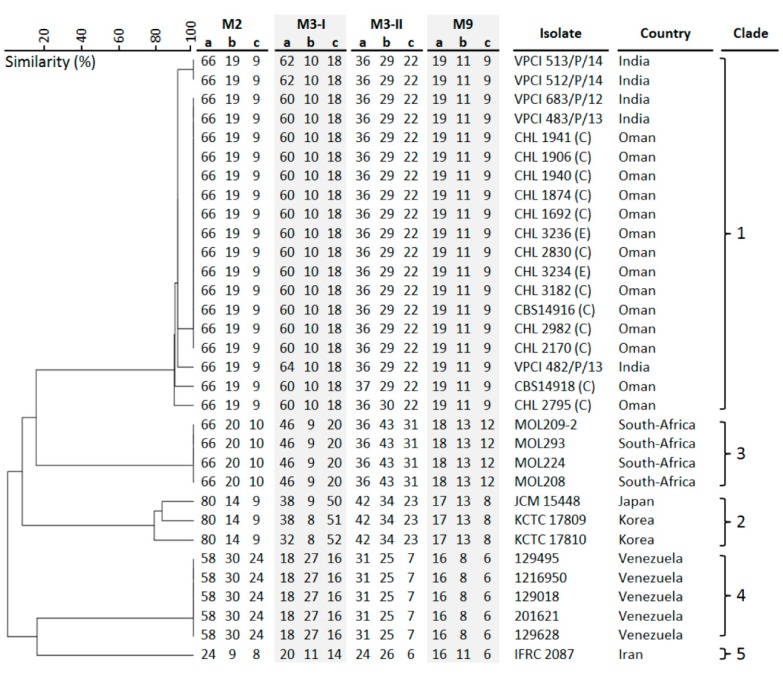
Small tandem repeat (STR) analysis of 12 STR *C. auris* targets with a repeat size of 2, 3, or 9 nucleotides. The isolates from Oman clustered with Indian isolates in the South Asian clade 1, while isolates from South Africa, Japan/Korea, Venezuela and Iran each clustered in the other 4 major *C. auris* clades. E = environmental; C = clinical.

**Table 1 jof-05-00101-t001:** Descriptive statistics of *Candida auris* cases in Sohar hospital (April 2018–April 2019).

Total Number	32
Median age in years (interquartile range)	64 (14–88)
Males (%)	17 (53.1)
Patients with comorbidities ^1^ (%)	24 (75)
Mean number of days to detection of *C. auris* ^2^ (SD)	28.4 (28)
Positive blood cultures for *C. auris* (%)	11 (34.4)
Positive urine cultures for *C. auris* (%)	14 (43.8)
Samples collected for screening (%)	7 (21.9)
Treated with antibiotics ^3^ (%)	16 (50)
Treated with antifungals ^4^ (%)	20 (62.5)
Mortality (%)	17 (53.1)

^1^ Including: diabetes, hypertension, cardiovascular, neurological, and immunodeficiency diseases; ^2^ This was calculated as collection date minus admission date; ^3^ Included: cephalosporin, piperacillin/tazobactam, and meropenem; ^4^ Included: fluconazole, voriconazole, and liposomal amphotericin B.

**Table 2 jof-05-00101-t002:** MIC values of 12 *Candida* strains from the outbreak.

Strain No	ITS Identification	MALDI-TOF MS Score	MICs Values of Clinical Isolates (mg/L)							
			AMB	FLC	ITC	VOR	POS	ISA	ANI	MICA
CHL 1941 (C)	*C. auris*	*C. auris* 2.13	1	>64	0.125	0.5	0.063	0.125	0.016	0.031
CHL 1906 (C)	*C. auris*	*C. auris* 2.21	2	>64	0.25	0.5	0.063	0.125	0.031	0.063
CHL 1940 (C)	*C. auris*	*C. auris* 2.30	1	>64	0.125	0.5	0.063	0.063	0.016	0.031
CHL 1874 (C)	*C. auris*	*C. auris* 2.16	1	>64	0.25	0.5	0.063	0.063	0.016	0.031
CHL 2170 (C)	*C. auris*	*C. auris* 2.31	1	16	<0.016	0.063	<0.016	<0.016	0.031	0.031
CHL 1692 (C)	*C. auris*	*C. auris* 1.92	2	>64	0.125	0.5	0.063	0.063	0.031	0.063
CHL 3236 (E)	*C. auris*	*C. auris* 2.22	1	8	<0.016	0.063	<0.016	<0.016	0.031	0.031
CHL 2830 (C)	*C. auris*	*C. auris* 2.28	2	>64	0.25	0.5	0.063	0.125	0.016	0.063
CHL 2982 (C)	*C. auris*	*C. auris* 2.39	1	8	<0.016	0.063	<0.16	<0.016	0.063	0.063
CHL 3234 (E)	*C. auris*	*C. auris* 2.19	2	16	<0.016	0.063	<0.016	<0.016	0.031	0.031
CHL 3182 (C)	*C. auris*	*C. auris* 2.17	1	>64	0.25	1	0.063	0.125	0.063	0.063
CHL 2795 (C)	*C. auris*	*C. auris* 2.20	1	16	<0.016	0.063	<0.016	<0.016	0.031	0.063

C: Clinical; E: Environmental; AMB = amphotericin B; FLC = fluconazole, ITC = itraconazole, VOR = voriconazole, POS = posaconazole, ISA = isavuconazole, ANI = anidulafungin, and MICA = micafungin.

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
