# Peer review of "Ongoing Challenges with Healthcare-Associated Candida auris Outbreaks in Oman"

_jof, 2019, doi:10.3390/jof5040101_

Round 1

Reviewer 1 Report

In this well-written manuscript clinical observations of a Candida auris outbreak at the Sohar hospital in Oman are presented. The structure is very clear, but some methodological details need to be refined before this manuscript can be published.

Major comments:

Information about the disinfection method used is confusing. In the material and methods section only hydrogen peroxide is mentioned (line 119). Later in the text (lines 271-272) the use of a “chlorine based disinfectant” is mentioned, and in the discussion section (lines 316-319) it is argued that the same protocol that in other reports was used, but those protocols included UV light which is not mention in the rest of the text. Since decontamination of C. auris from hospital wards is one of the most worrisome issues, this point must be carefully and consistently explained.

The Statistical Analysis section (lines 158-160) lacks details, e.g. software versioning, and more importantly background information on the regression analysis done in R (ideally the authors should provide the R scripts in the supplement).

The subheadings “laboratory methods “ (line 128), “mycology” (line 208) and “molecular identification” (line 221) should be changed to something with more explanatory power or a concluding statement.

Please note, that the ITS (Internal Transcribed Spacer) region is NOT a gene (lines 222-223).

Formatting: Missing (especially between absolute numbers and percentages in parentheses) and extra spaces throughout the text need to be corrected.

Specific comments:

Lines 41 and 42: …patient’s… is incorrect, since it was not a single patient who had been sampled, this needs to be …patients’…

Line 108-111: Very few information about the procedures followed has been included. More details, or references if applicable, must be included.

Line 115-117: all items in list need to be in plural.

Line 132-133: Was the Bruker and BioMerieux analysis already run against existing C. auris profiles? (see also comment line 217)

Line 134-135: Where in the Netherlands? Details about sequencing and molecular typing are missing.

Line 137-139: Further details for DNA extraction are needed. Glass beads by themselves are not an extraction method. Appropriate reference(s) should be included here.

Line 139-140: Further details for PCR amplification are missing: oligonucleotide sequences, enzyme used, program, etc. and provide an appropriate reference.

Line 140-141: Details about sequencing, method or service, need to be included. What does “DNA sequences were edited” mean?

Line 148-151: Referring to a method as “unpublished” is wholly unsatisfactory and utterly inappropriate, this needs to be explained in full here.

Table 1 (Line 175): What does C/s stand for?

Line 197: Definition of the abbreviation MERS-CoV would be helpful for non-clinicians.

Line 198: Define the “enhancement of infection control”.

Line 209: “32 were found to be positive” of what? Please rephrase sentence to clarify.

Line 217: Is the “in-house database” somewhere available or has it been previously published? It would be hard to judge the validity of these results without further details.

Line 222: Start sentence with: “A part of the ITS region was used …”

Line 223: “enough polymorphism” (should be “enough polymorphisms”) is a vague statement, could this statement be firmed up with a few numbers, and provide reference for this.

Line 227: The text argues that “Candida species within the C. haemulonii complex were selected for phylogenetic analyses” but in the phylogenetic tree (Fig. 3) more Candida species are included. This point must be clarified.

Line 231: The sentence referring to Figure 4 does not match with what is shown in Figure 4.

Line 233: Include “whole-genome sequencing (WGS)” instead of just “WGS”, this also requires reference(s) about the “5 different C. auris clades”.

Line 246-250: Are these MIC50 values? Please specify! Also add a few sentences about how it was decided whether certain MICs for particular antifungals are considered low or high.

Line 252 (Table 3): Antifungal acronyms must be defined in the table footnotes.

Line 261-262: better say “…training all staff of involved units on …”

Line 265: sentence is not finished at fullstop after “staff”.

Line 299: define CVCs.

Author Response

Reviewer: 1

In this well-written manuscript clinical observations of a Candida auris outbreak at the Sohar hospital in Oman are presented. The structure is very clear, but some methodological details need to be refined before this manuscript can be published.

Major comments:

Information about the disinfection method used is confusing. In the material and methods section only hydrogen peroxide is mentioned (line 119). Later in the text (lines 271-272) the use of a “chlorine based disinfectant” is mentioned, and in the discussion section (lines 316-319) it is argued that the same protocol that in other reports was used, but those protocols included UV light which is not mention in the rest of the text. Since decontamination of C. auris from hospital wards is one of the most worrisome issues, this point must be carefully and consistently explained.

Response 1: clarified in the revised submission

The Statistical Analysis section (lines 158-160) lacks details, e.g. software versioning, and more importantly background information on the regression analysis done in R (ideally the authors should provide the R scripts in the supplement).

           Response 2: Table 2 was removed as suggested by reviewer 2

The subheadings “laboratory methods “ (line 128), “mycology” (line 208) and “molecular identification” (line 221) should be changed to something with more explanatory power or a concluding statement.

           Response 3: Changed

Please note, that the ITS (Internal Transcribed Spacer) region is NOT a gene (lines 222-223).

           Response 4: corrected

Formatting: Missing (especially between absolute numbers and percentages in parentheses) and extra spaces throughout the text need to be corrected.

           Response 5: Done

Specific comments:

Lines 41 and 42: …patient’s… is incorrect, since it was not a single patient who had been sampled; this needs to be …patients’…

Response 6: Corrected

Line 108-111: Very few information about the procedures followed has been included. More details, or references if applicable, must be included.

Response 7: details added

Line 115-117: all items in list need to be in plural.

Response 8: Corrected

Line 132-133: Was the Bruker and BioMerieux analysis already run against existing C. auris profiles? (see also comment line 217)

Response 9: Yes, using Bruker all strains were C. auris whereas using BioMerieux & Vitek 2, we got different Candida species such as Candida haemuloniiCandida duobushaemuloniiRhodotorula glutinis and  Candida sake. All our Candida strains were further identified by sequencing and maldi-tof.

Line 134-135: Where in the Netherlands? Details about sequencing and molecular typing are missing.

Response 10: We have added some details.

Line 137-139: Further details for DNA extraction are needed. Glass beads by themselves are not an extraction method. Appropriate reference(s) should be included here.

Response 11: DNA extraction protocol details and appropriate reference were added.

Line 139-140: Further details for PCR amplification are missing: oligonucleotide sequences, enzyme used, program, etc. and provide an appropriate reference.

Response 12: Details for PCR amplification and sequencing were added.

Line 140-141: Details about sequencing, method or service, need to be included. What does “DNA sequences were edited” mean?

Response 13: Details for sequencing were added. We mean that the sequences were checked for the following: 1. Check the chromatogram of sequencing results and quality; 2. Edit the basecalling reads manually 3. Automatically align forward and reverse reads

Line 148-151: Referring to a method as “unpublished” is wholly unsatisfactory and utterly inappropriate; this needs to be explained in full here.

Response 14: We have extended the methods section and provided a reference on development of the STR. New Ref De Groot et al. 2019 BioRXiv.

Table 1 (Line 175): What does C/s stand for?

Response 15: C/s stands for cultures; abbreviation was removed and full word inserted.

Line 197: Definition of the abbreviation MERS-CoV would be helpful for non-clinicians.

Response 16: Agreed and changed.

Line 198: Define the “enhancement of infection control”.

Response 17: More clarification was added.

Line 209: “32 were found to be positive” of what? Please rephrase sentence to clarify.

Response 18: Done

Line 217: Is the “in-house database” somewhere available or has it been previously published? It would be hard to judge the validity of these results without further details.

Response 19: the sentence “in house data base” has been removed because the newest version of the commercial Bruker database (Maldi Biotyper MBT Smart V8.0.0.0.7171) detects C. auris from 4 clades.

Line 222: Start sentence with: “A part of the ITS region was used …”

Response 20: Done

Line 223: “enough polymorphism” (should be “enough polymorphisms”) is a vague statement, could this statement be firmed up with a few numbers, and provide reference for this.

Response 21: Corrected

Line 227: The text argues that “Candida species within the C. haemulonii complex were selected for phylogenetic analyses” but in the phylogenetic tree (Fig. 3) more Candida species are included. This point must be clarified.

Response 22: Corrected

Line 231: The sentence referring to Figure 4 does not match with what is shown in Figure 4.

Response 23: We have updated this sentence to make it more clear. Figure 4 is also expanded to show all 5 clades.

Line 233: Include “whole-genome sequencing (WGS)” instead of just “WGS”, this also requires reference(s) about the “5 different C. auris clades”.

Response 24: reference 4 and 44 now mentioned here

Line 246-250: Are these MIC50 values? Please specify! Also add a few sentences about how it was decided whether certain MICs for particular antifungals are considered low or high.

Response 25: No, all MICs in the table are individual MICs. We included a sentence to show that we related low and high MICs to published ECOFF values of C auris with a reference [42]. Therefore we changed low to “below the ECOFF” and high MIC was changed to nonsusceptible (>ECOFF). For fluconazole and amphotericin B we included the MIC 50 and 90.

Line 252 (Table 3): Antifungal acronyms must be defined in the table footnotes.

Response 26: Defined

Line 261-262: better say “…training all staff of involved units on …”

Response 27: Corrected

Line 265: sentence is not finished at full stop after “staff”.

Response 28: Corrected

Line 299: define CVCs.

Response 29: done

Reviewer 2 Report

The authors reported data about a C. auris outbreak in a single Hospital in Oman. The data are interesting and more information from all over the world about outbreak are still needed. However, I have the following comments:

ABSTRACT

1) Spell out ICU, CCU some readers my be not familiar with these terms

2) Line 49, as I will state below, I disagree on reporting mortality rate in patients with "comorbidities" because this term is too generic. I suggest to delete this and insert only a range of crude mortality

INTRODUCTION

1) LINE 64 I suggest the author to add the following reference to support the global trend the authors described: doi: 10.1186/s40560-018-0342-4

2) I suggest the authors to close their introduction with the aim of their paper. From Line 86-88 can be moved to the discussion

METHODS

1) Line 92: This hospital has a "level of care"? Just to have an idea of the volume of the hospital

2) LINE 104-106 Use was instead of is

RESULTS:

1) LINE 163: I suggest to separate the timeline and the description of patients' cohort

2) line 172: Which class of antibaterial drugs? Insert this information if available

3) Line 189-193: I do not think that this univariate analysis add much to descriptive statistics. The reason is that some variables are too generic (i.e. comorbidities) and because, in the univariate analysis, there can be overlap of associations. So, I suggest the authors to delete it.

DISCUSSION

1) line 285: You discuss this in the introduction. I suggest to start with discussion of the main findings of your reports

2) line 296-298: As I write before, I would not rely on comorbidities as a "whole". So I would not cite the results of the univariate analysis

3) line 299-304: I would add some words to discuss why the ICU is an hotspot for C. auris. The reason is the vicious cycle of environment/workers contamiantion - patient contamination - infection which is very had to break. Consider adding this reference for this concept: doi: 10.1186/s13054-019-2449-y

4) From 299 to 312: ok this discussion (with some add - see comment above) but what is the link with your data?

5) A comparison with the data from previous reports of cases in Oman would be interesting doi: 10.1111/myc.12647 ; doi: 10.1016/j.jinf.2017.05.016

5) line 334-338: I would change this last part with a conclusion specifically about your data on the outbreak in Oman

Author Response

Reviewer: 2

The authors reported data about a C. auris outbreak in a single Hospital in Oman. The data are interesting and more information from all over the world about outbreak are still needed. However, I have the following comments:

ABSTRACT

Spell out ICU, CCU some readers maybe not familiar with these terms

Response 1: Done

Line 49, as I will state below, I disagree on reporting mortality rate in patients with "comorbidities" because this term is too generic. I suggest to delete this and insert only a range of crude mortality

Response 2: Agree, Done

INTRODUCTION

Line 64, I suggest the author to add the following reference to support the global trend the authors described: doi: 10.1186/s40560-018-0342-4

Response 3: We have added the following reference; Cortegiani et al. to the manuscript [3].

I suggest the authors to close their introduction with the aim of their paper. From Line 86-88 can be moved to the discussion

Response 4: Done

METHODS

Line 92, This hospital has a "level of care"? Just to have an idea of the volume of the hospital

Response 5: Added

Line 104-106, Use was instead of is

Response 6: Corrected

RESULTS:

Line 163, I suggest to separate the timeline and the description of patients' cohort

Response 7: Done

Line 172, Which class of antibaterial drugs? Insert this information if available

Response 8: This was mentioned in the legend of table 1

Line 189-193, I do not think that this univariate analysis add much to descriptive statistics. The reason is that some variables are too generic (i.e. comorbidities) and because, in the univariate analysis, there can be overlap of associations. So, I suggest the authors to delete it.

Response 9: Agreed and it was deleted

DISCUSSION

Line 285, You discuss this in the introduction. I suggest to start with discussion of the main findings of your reports

Response 10: Corrected

Line 296-298, As I write before, I would not rely on comorbidities as a "whole". So I would not cite the results of the univariate analysis

Response 11: Agree and corrected

Line 299-304, I would add some words to discuss why the ICU is a hotspot for C. auris. The reason is the vicious cycle of environment/workers contamiantion - patient contamination - infection which is very hard to break. Consider adding this reference for this concept: doi: 10.1186/s13054-019-2449-y

Response 12: Done.

From 299 to 312, ok this discussion (with some add - see comment above) but what is the link with your data?

Response 13: Modified.

A comparison with the data from previous reports of cases in Oman would be interesting doi: 10.1111/myc.12647 ; doi: 10.1016/j.jinf.2017.05.016

Response 14: Done.

line 334-338: I would change this last part with a conclusion specifically about your data on the outbreak in Oman

Response 15: Done

Round 2

Reviewer 2 Report

I have no further comments.